# Exploring fertility preservation in AYA cancer survivors: Information needs and post-cancer challenges

**Manon Vialle**[1], **Jacqueline Saias-Magnan**[2], **Anne Dezamis**[3], **Catherine Guillemain**[2,4], **Anne-Déborah Bouhnik**[5], **Julien Mancini**[5,6], **Blandine Courbiere**[2,7*]

**1** Centre Norbert Elias (UMR 8562), Centre National de la Recherche Scientifique (CNRS), Marseille, France, **2** Hôpital de la Conception – AP-HM, Department of Reproductive Medicine-CECOS, Pôle Femmes-Parents-Enfants/ Plateforme Cancer & Fertilité ONCOPaca-Corse, Marseille, France, **3** Institut Paoli Calmette, Department of Clinical Psychology, Marseille, France, **4** Aix Marseille Univ, INSERM, MMG, UMR_S1251, Marseille, France, **5** Aix Marseille Univ, INSERM, IRD, SESSTIM, Sciences Économiques et Sociales de la Santé & Traitement de l'Information Médicale, ISSPAM, Marseille, France, **6** APHM, BioSTIC, Biostatistique et Technologies de l'Information et de la Communication, Marseille, France, **7** Aix Marseille Univ, Avignon Univ, CNRS, IRD, IMBE, Marseille, France

* blandine.courbiere@ap-hm.fr

## Abstract

### Purpose

To explore the experiences of fertility preservation (FP) and cancer-induced infertility among adolescent and young adult (AYA) cancer survivors.

### Methods

This cross-sectional, qualitative study was conducted by a sociologist in collaboration with a multidisciplinary team specializing in oncology and reproductive medicine. The sample included 31 AYA cancer survivors (13–35 years old) in remission for at least one year. Participants were recruited from oncology or reproductive departments and via "snowball" sampling. Semi-structured interviews, lasting 1–3 hours, were conducted.

### Results

Of the 31 participants (18 women, 13 men), 12 women and 12 men had undergone FP, while 6 women and 1 man had not. Some were fertile, some infertile, and others were unsure of their fertility status. Six women and 5 men had children post-cancer, while others were planning or not considering parenthood. Thus, the respondents' experiences of fertility were varied, and their analysis led to a number of observations: 1) a lack of attention to infertility risks prior to cancer treatment; 2) fertility preservation not linked to an imminent pregnancy after cancer raises unconsidered difficulties; 3) gaps in information and care beyond the question of fertility.

**Data availability statement:** We understand the importance of data sharing in ensuring transparency and replicability in research. However, in this case, we have opted for a restriction due to the sensitive nature of the data and our ethical commitments to the research participants. The raw data in our study consist of qualitative interviews that, even when pseudonymized, contain personal narratives that could compromise participant confidentiality. As part of our ethical protocol approved by the ethics committee, we guaranteed participants anonymity and the confidentiality of their data. Additionally, the protocol mandates the destruction of the interviews 5 years after analysis to further protect participant privacy. Here are the contact details of the Aix-Marseille University ethics committee to which data requests can be sent. This is the committee that validated the ethical and methodological protocol for this research. Contact: audrey.janssens@univ-amu.fr. Although we are unable to share the raw interview data, we believe that the methodology and tools used ensure the reproducibility of our study. To this end, we have expanded on the presentation of the interview guide in the article (page 6) and the thematic analysis of the interviews (page 9). This will provide other researchers with the information they need to understand and, if necessary, reproduce our methodology and compare their results.

**Funding:** MV INCA SHS-E-SP N°2017-137 The Institut National du Cancer (INCa) URL: https://www.e-cancer.fr/Institut-national-du-cancer/Appels-a-projets/Appels-a-projets-resultats/SHS-RISP-2024.The funders had no role in study design, data collection and analysis, decision to publish, or preparation of the manuscript.

**Competing interests:** The authors have declared that no competing interests exist.

## Conclusions

The study reveals a lack of information and attention to AYA-specific issues regarding fertility and endocrine function after cancer. There is a need for tailored informational resources for AYA survivors.

## Introduction

The rising incidence of cancer diagnoses among adolescents and young adults (AYA) over recent decades, coupled with advancements in treatment, has led to an increase in the number of AYA cancer survivors [1]. As a result, there is a growing shift in the medical approach, extending beyond the treatment phase to encompass the post-cancer period [2]. Over the past decade, numerous studies have documented the long-term consequences of cancer and its treatments, which often affect the quality of life in both physiological and psychological dimensions. These consequences may impact fertility, sexuality, relationships, the desire for children, professional life, and more [3–10].

In particular, research on infertility risks has highlighted the importance of preserving the possibility of parenthood after cancer, which plays a significant role in the process of self-reconstruction and reintegration into "normal" life [11–12]. This goal remains achievable despite the gonadotoxic effects of chemotherapy, thanks to advancements in fertility preservation and assisted reproduction techniques, which continue to offer new hope—even for prepubescent individuals [13–15]. In women, fertility preservation (FP) techniques, such as the cryopreservation of oocytes, embryos, ovarian tissue, and the ovarian tissue transplantation, have demonstrated promising live birth rates [16–17]. Consequently, the systematic offer of fertility preservation prior to cancer treatment is now worldwide recommended by all scientific societies [18–20].

However, many studies [5,11,21–24] have consistently reported that post-cancer challenges, including those related to sexuality, fertility, and relationships, are often overlooked in medical care. Advice on the risks of cancer-induced infertility and the availability of fertility preservation is not always provided during oncology treatment [5,25]. As a result, patients often report a lack of knowledge on these issues, emphasizing the need for better information and support throughout and after cancer treatment [11,12,22,24,26–28]. This knowledge gap continues to generate stress and anxiety for patients regarding the potential impact of cancer treatments on fertility [26–27].

At the same time, some oncology units have organized individualized care pathways for the AYA population, not only focusing on curative treatments but also addressing psychosocial support [29]. In collaboration with fertility specialists, these units aim to address AYA survivors' future reproductive needs by offering systematic fertility preservation services [30].

As a result, AYA cancer survivors now face a range of different situations [31]. Some have undergone fertility preservation, while others have not. For those

considering parenthood, some will need to use preserved gametes or germinal tissue through assisted reproductive technologies (ART), while others may rely on donated gametes. Additionally, some survivors who remain fertile may conceive "spontaneously," regardless of whether they underwent fertility preservation.

This study, based on a qualitative sociological survey, aims to explore AYA survivors' post-cancer experiences in three key scenarios: 1) when the risk of infertility was not anticipated and fertility preservation was not offered; 2) when fertility preservation was undertaken but the AYA survivors do not yet have plans for parenthood; and 3) when fertility preservation was implemented as part of an ongoing parental project.

## Materials and methods

### Study population

This research was conducted in 2018 and 2019 as part of a cross-sectional, qualitative study led by a sociologist in collaboration with a multidisciplinary team of physicians specializing in oncofertility, in partnership with an oncology unit.

The inclusion criteria targeted women and men who were adolescents and young adults (AYA) at the time of their cancer diagnosis, who had been in remission for at least one year and who have reached the age of majority at the time of interview. The AYA category was defined broadly, ranging from 13 to 35 years old, in line with recent studies—some of which begin this category at age 13 [32], while others extend it up to 39 years old [33–37]. This wide age range was chosen to form a relevant cohort, particularly considering the growing trend of delayed parenthood [38], and to include individuals who did not have children at the time of their cancer diagnosis, for whom fertility becomes an even more pressing concern after treatment.

Three cancer types were selected for inclusion: breast cancer, testicular cancer, and malignant hematological diseases. We focused on three types of cancer in order to maintain a certain level of homogeneity in both the types of cancers and the oncology and oncofertility care pathways, allowing us to better identify the specificities arising within these different journeys. We chose to concentrate on hematologic cancers, which particularly affect young people, as well as two more gender-specific cancers—testicular cancer and breast cancer—one of which (breast cancer) can be hormone-dependent in women, in order to take into account how gender and hormonal factors may influence the experience of illness, fertility preservation, and reproductive decisions.

Participants were recruited between 20 November 2018 and 9 December 2019 through various channels, including oncology and reproductive medicine departments, snowball sampling, and outreach via public testimony calls. In the medical reproduction department, an on-site presence, combined with the help of the department's secretaries and doctors, enabled this research to be systematically presented to eligible patients, who were systematically offered an interview, to be conducted at a later date if they so wished. In the oncology department, in partnership with the AYA department, an email presenting the research project and a call for participation were sent out to their email database. These two recruitment sources were the main ones, and in addition, word of mouth and the snowball effect helped expand recruitment beyond the locality of the two main departments, allowing for the consideration of any local specificities in the care provided. These recruitment methods and channels combined ensured a greater diversity of respondent profiles in terms of place of residence and care.

In-depth, semi-structured interviews were mainly conducted face-to-face, or exceptionally by videoconference or telephone. Interviews lasted between 1 and 3 hours.

The first part of the interview guide focused on sociodemographic questions (age, gender, marital status, education, profession). The next sections were structured chronologically. A second part addressed the participants' personal, marital, and professional history up until the cancer diagnosis, as well as the possible emergence of a desire for a child during this period. The third part dealt with the cancer diagnosis, its treatment, experiences with medical care, including fertility-related issues, as well as the impacts and concerns in professional, social, family, and marital life. The fourth part of the interview focused on experiences after the completion of cancer treatment. Questions revisited the potential

consequences of cancer and its treatments afterward, regarding fertility status (whether gamete preservation was performed or not, and whether fertility was preserved after treatment, if that information was known), and the experiences related to these aspects. Other questions then addressed the marital history, professional situation, desire for children after the end of treatment, and any parental project that was being considered, underway, or completed.

All interviews were audio-recorded, transcribed verbatim, and anonymized to protect participants' privacy.

## Analysis

The interviews were analyzed using an inductive approach, commonly referred to as grounded theory [39]. We employed a combination of social science methods, including longitudinal interview analysis and cross-sectional analysis. Longitudinal interview analysis focuses on identifying the individual processes and organizational patterns within each interview. In contrast, cross-sectional analysis examines the interviews across participants to identify common themes. These two analytical methods are complementary; together, they allow us to capture both the unique aspects of individual narratives and the broader, shared themes across respondents' experiences. By combining these methods, we were able to highlight both the similarities and differences in the participants' discourses, while also respecting the distinctiveness of each participant's account.

## Ethical approval

This study was approved by the ethics committee of Aix-Marseille University (France) on November 13, 2018 (Approval Reference: 2018-08-11-011). A written information sheet was provided for each participant to read, and was also explained orally. A written consent form was also distributed and explained orally. The persons contacted had to read it, approve each of its clauses and sign it before participating in the study. The consent form to be signed specified that the person had read and understood the information sheet presenting the research and its procedure, that he/she agreed to participate, that the interview would be recorded, and that he/she understood that he/she could interrupt the recording or the interview at any time without any consequences, not to answer any questions if he/she did not wish to do so, that the data were anonymous and protected by the law on the protection of individuals and liberties, and that he/she could contact the investigator at any time to ask questions or withdraw from the study if he/she so wished. All the data collected was kept by the research investigator on a secure password-protected area. The data stored was all anonymized, and the ethical protocol approved by the committee stipulates that it must be destroyed 5 years after the end of the study. The data presented here have been pseudonymized, and all the first names appearing in the results section are fictional.

## Results

A total of 31 interviews were conducted with 18 women and 13 men who were adolescents or young adults (AYA) at the time of their cancer diagnosis and had been in remission for at least one year.

Three cancer types were included in the study: breast cancer (8 women), testicular cancer (5 men), and malignant hematological diseases (14 women, 8 men).

The women were aged 13–35 years at the time of their cancer diagnosis, with an average age of 28.5 years. The men were aged 16–29 years at diagnosis, with an average age of 24 years.

For women, the time since cancer treatment ranged from 1.5 to 18 years, with an average of 5.8 years and a median of 5 years. For men, the time since treatment ranged from 1 to 21 years, with an average of 4.3 years and a median of 3 years.

The participants' fertility statuses varied: some considered themselves still fertile (based on medical exams such as spermogram or following spontaneous pregnancy), others considered themselves infertile (confirmed by medical tests such as spermogram or due to the absence of menstruation), and some did not know whether they were fertile or not (due to the absence of various physiological signals – such as the absence of menstruation, or medical examination or attempts to conceive). Of the participants, 12 women and 12 men had undergone fertility preservation, while 6 women and

1 man had not (see Table 1). Some conceived after cancer (6 women, 5 men), others were trying to have children, while some had no plans for parenthood.

Among the themes emerging from the interviews conducted, we find discourse on the proposal for fertility preservation (whether it was offered or not, when, by whom and how); the experience of fertility preservation when it was performed; the aftermath of treatment following the fight against cancer; the short-, medium-, and long-term consequences of treatments; the ignorance and late discovery of the effects on fertility; the misunderstanding of post-cancer difficulties on the part of relatives; the discovery of the premature ovarian insufficiency for women; experiences of infertility, whether real or assumed; the lack of information on the use of cryopreserved gametes and post-cancer infertility; the desire for children or not; the parental project; and the difficulties faced in assisted reproduction treatments (ART). Within each of these themes, several other categories emerged. In this paper, we will focus on certain data and concentrate our analysis on the following points: 1) the lack of attention to infertility risks prior to cancer treatment; 2) fertility preservation and the specific challenges it presents when not linked to a parental project after cancer; and 3) gaps in information and care beyond the question of fertility, often considered only in the context of medically assisted reproduction.

## 1. Lack of attention regarding infertility risks before cancer treatment

Not all respondents had the opportunity to preserve their fertility before undergoing cancer treatment [Table 1]. This variation was influenced by both gender and the timing of their cancer diagnosis. For instance, among the men interviewed, the only participant who did not benefit from fertility preservation (FP) had been diagnosed with lymphoma 22 years earlier, in 1997, when fertility preservation was not yet a medical or social concern. In contrast, the 12 other men, all diagnosed more recently (between 2014 and 2018), had access to FP. We will not explore further the experience of this one man participant, who was aware of his infertility from the outset, understood the reasons for the lack of FP at the time, and did not express particular concern during the interview. In addition, the greater number of men than women using FP in our sample [Table 1] can be explained by a point that is well documented in the literature, namely that gamete retrieval is easier for men than for women in terms of time, technique and invasiveness for the body [40]. As regards the use of FP by the women in our sample, the situations are indeed more varied and complex.

Among the 18 women interviewed, 10 had undergone FP at the time of their cancer diagnosis (between 2009 and 2017). Among the 8 women who had not had FP (diagnosed between 2001 and 2018), 6 were never offered the option, 1 attempted FP unsuccessfully, and 1 did not undergo chemotherapy or radiotherapy (only surgery) and was not offered FP for that reason. Interestingly, two of the 6 women who had not been offered FP were later able to preserve their fertility post-treatment.

**Table 1. Participant's cancer type, fertility and parenthood situation.**

| | | Number of Women | Number of Men |
|---|---|---|---|
| Number of participants | | 18 | 13 |
| Cancer Type | Breast Cancer | 7 | / |
| | Malignant Hematological Diseases (Leukemia, Hodgkin's and Non-Hodgkin's Lymphoma) | 11 | 8 |
| | Testicles Cancer | / | 5 |
| Fertility Preservation (FP) | FP Before Cancer Treatment | 10 | 12 |
| | FP After Cancer Treatment | 2 | / |
| | No FP | 6 | 1 |
| Child After Cancer | Spontaneous Conception | 3 | 1 |
| | ART With Preserved Gametes | 1 | 3 |
| | ART With Donated Gametes | 2 | 1 |
| | Childless | 12 | 8 |

In this section, we focus on the experiences of the 6 women who were not given the option of FP at the time of their diagnosis. The aim is to better understand how they experienced the absence of this option and the impact it had later on.

Of these 6 women, 2 were diagnosed before 2011 (in 2001 and 2004), at a time when oocyte vitrification was not yet allowed in France. Since 2011, the authorization of oocyte vitrification has significantly increased the availability of FP for women facing cancer, especially single women who could not benefit from embryo cryopreservation. These two women had the earliest cancer diagnoses, with Elina being the youngest, diagnosed at 13 years old. The other woman was 24 years old at the time of diagnosis. The absence of FP options for them was primarily due to the timing of their diagnoses and the limited awareness and technical options available in France at the time.

When discussing infertility in the interviews, these women expressed strong regret not only over the lack of FP but also over their initial ignorance of the effects of cancer and its treatment on their fertility. Elina, now 27, explained:

*"The early menopause thing, I knew about it right there, in August. I never knew before. But it wasn't a big surprise... You think that if chemotherapy defeats cancer, it must affect other parts of the body. So when the doctor told me, it was a surprise in that I didn't expect it this year, but it wasn't... either. But it's still a consequence, and it's thirteen years later. It's not a consequence that happens thirteen years later; it's something that was caused from the beginning, but I only learned about it thirteen years later."*

The four other women who had not been offered FP and were diagnosed after 2011 were unclear about the reasons for this oversight, except for Aïssa, who was initially treated in Algeria. She was not surprised by the lack of information, noting that there was little communication or education regarding cancer treatments in the Algerian department where she was treated. The three other women, treated in France at ages 28, 31, and 34, were childless at diagnosis and wanted to have children. They were also left without an explanation for the lack of fertility preservation. What they lamented most was the absence of information on the risk of infertility after treatment. Martine, 34 years old at the time of her leukemia diagnosis in 2012, now 40 and childless, shared:

*"It's been pathetic. Now, really, it's... I hope it's better now. I was never told that chemotherapy would make me sterile, that it might cause infertility either because of the treatment itself or the chemotherapy – I had zero information, nothing at all."*

All 6 women who were not offered FP had one thing in common: they were not informed about the infertility risks linked to their cancer treatments, and this late realization left them ill-prepared. For some, like Elina, this information came several years after treatment, at a time when they were hoping to move beyond the cancer experience. The need for fertility assessments, subsequent treatments for preservation, or the discovery of infertility became an additional and unpleasant challenge that could have been anticipated.

Women who had access to FP before their cancer treatments were spared these post-cancer medical procedures (such as ovarian reserve assessments, ovarian stimulation, or oocyte retrieval) and the anxiety surrounding potential infertility. However, the two women who were able to undergo fertility preservation after treatment found this option to be both highly beneficial and restorative, as it addressed an important unmet need. These two cases, while exceptional within this study, underscore the importance of offering fertility preservation post-treatment when the individual's fertility is still viable and when it was not possible beforehand.

## 2. Fertility preservation independently of a parental project after cancer: Unintended consequences

Among the 31 individuals interviewed, ten women and twelve men were able to undergo fertility preservation (FP) during their cancer treatment. All of the men had sperm freezing, while the women used a variety of techniques: five had oocyte

 

vitrification (one of which involved immature oocytes), three had ovarian tissue cryopreservation, and two had embryo cryopreservation.

Of the ten women who underwent fertility preservation, five became mothers after treatment: three through spontaneous pregnancy, one via IVF with egg donation (because the post-chemotherapy cryopreserved ovarian tissue contained no follicles), and one through IVF with their frozen-thawed oocytes. One other woman was pregnant at the time of interview, after IVF. Among the four women who were childless and not pregnant, two were undergoing IVF for infertility, and two were single (aged 29 and 32) with no current desire for children.

Among the twelve men who underwent FP, one has had a child after IVF, three were expecting a child (two after IVF with cryopreserved sperm and one after spontaneous pregnancy), two were trying to conceive with their partner (under ART), and six, aged between 22 and 30 and single, were not planning to have children at the time of the interview.

The experiences of these women and men with fertility preservation varied hugely. Some considered FP as a positive and empowering step that helped them move forward in their post-cancer lives. However, others experienced it as a source of new challenges and questions. This section focuses on the latter group. Our findings suggest that these more negative experiences are closely linked to the question of whether a parental project is a current or future project. The respondents who were not involved in a parental project and were single (two women and six men, aged between 22 and 32) expressed the most concern. Their testimonies can be gathered into three main themes: a/ The challenge posed by a known infertility and the need for ART in future relationships; b/ Complain about the annual letter from the fertility clinic, following cryopreservation, which asks young, single individuals without children to decide the fate of their cryopreserved gametes—an "impossible choice" for them; c/ The lack of long-term counseling regarding fertility after cancer, which leads to uncertainty and, at times, misconceptions, resulting in unnecessary stress.

### a. The challenge posed by assumed infertility and the need for Assisted Reproductive Technologies (ART) in future relationships.

*"You have to know that you need to find someone who's open-minded enough because it won't happen naturally; it will have to be through injections… not something that happens naturally, so you need to find someone who's okay with that. You have to discuss it with them beforehand, and that can really slow things down."* (Rémy, 24, testicular cancer at 20, single with no children, sperm cryopreserved pre-chemotherapy, with sexual dysfunction and no ejaculation)

*"It raises a lot of questions, as I've said. Even though I know I'm infertile, even with preservation, I'll never have children of my own blood. And it's true that when I was told my sperm were altered, we couldn't know if they'd be viable until they were thawed. That made me doubt—what if I can't have biological children? I'd have to adopt, but adoption is still really complicated. So it makes you question everything, including meeting someone who's told they can't have children. It's always delicate to bring that up. You never know if it will change things overnight."* (Amine, 26, Hodgkin's lymphoma at age 21, single with no children, pre-chemotherapy sperm cryobanked but cryopreserved sperm of poor quality, azoospermia after cancer)

*"For me, it's stressful. The day I want to have a child, if I have no choice but to go through ART, I'll have to go to the lab for that... And that stays in my mind. Today, I'm single, and I think to myself, 'If I meet someone, and we have to go through this, it's...'. In terms of future relationships, well, at least for me, I've always felt guilty about imposing this on someone."* (Florie, 37, hormone-dependent breast cancer at age 31, single with no children, no FP before but after treatment, premature ovarian insufficiency (POI) after treatment)

These excerpts highlight the concerns faced by single respondents who were not engaged in a parental project at the time of the interview. They anticipate potential difficulties in forming relationships due to their infertility and the possible need for ART in the future. The uncertainty about the true state of their fertility, the quality of their cryopreserved gametes, and the

implications for their future family plans leads them to speculate about a potentially challenging path to parenthood. This uncertainty also shapes their thoughts about the impact of infertility on entering a relationship with a partner who desires children.

### b. The annual letter from the ART laboratory: Impossible choices for single AYAs without children?

When fertility preservation has been carried out, the French Law requires that the ART laboratory sends an annual letter to the gamete owner, presenting four options: continue preserving the gametes for personal use, donate the gametes to others, donate them for scientific research, or destroy them. This letter, along with the difficult decision it requires, was frequently discussed by the younger respondents.

*"It's a letter I receive once a year, and I've been waiting for it, well, for two years now. After the third one, I was like, 'Oh yes, they really send it every year.' So I thought, 'Well, no, I'm just going to keep them.' And since I know the date they send it, this year I was like, 'Well, I haven't received it yet.' And indeed, it came a week later—slightly delayed—but I got it. So yeah, it's kind of something I'm waiting for. When I have it in my hands, I don't know what to do with it, so I check the box to delay the decision, but I'm still waiting for it every year."* (Thibault, 22, Hodgkin's lymphoma at 16, single with no children, sperm cryopreserved pre-chemotherapy, normal spermogram post-cancer)

Testimonies like Thibault's reveal that this letter arrived at a time when the individuals were not planning to have children, were unsure if they will want children in the future, and were uncertain whether they will need these cryopreserved gametes. This generates high confusion and anxiety over what should happen to the cryopreserved gametes when their personal use is still uncertain. Furthermore, some respondents reported feeling uncomfortable receiving this letter, as it forces them to revisit their cancer experience at a time when they would prefer to distance themselves from the medical aspects of their past.

### c. Lack of long-term counseling about fertility preservation after cancer leads to uncertainty and misconceptions.

Some respondents assumed there would be future financial costs associated with the ongoing cryopreservation of their gametes, despite the fact that in France, public national-health insurance covers all costs related to FP:

*"And do you receive a letter every year about the frozen sperm?*

*Yes, I do. I think you have to pay, right? Or is the first year free? Something like that. I think you have to pay."* (Rémy, 24, testicular cancer at 20, with sexual dysfunction and no ejaculation, single with no children, sperm cryopreserved pre-chemotherapy)

*"I know I won't be able to keep them forever. During the five years of remission, it's free, and that's fine, I'm covered. But I know that after that, I'll have to pay to keep my sperm. Once I'm out of remission, I won't be covered anymore, and I'll need to pay the ART lab to keep my sperm. So, keeping them for five years was just a fallback option. When I checked the box to keep them, I didn't know whether to destroy or donate them."* (Thibault, 22, Hodgkin's lymphoma at 16, single with no children, sperm cryopreserved pre-chemotherapy, normal spermogram post-cancer)

In France, a cancer diagnosis entitles patients to full coverage of treatment costs related to the disease, including fertility preservation, for up to five years after the end of treatment. However, uncertainty about the costs and conditions of gamete preservation—particularly beyond the remission period—led some respondents, like Thibault, to hastily question what should happen to their cryopreserved gametes. This uncertainty was compounded by concerns about the potential financial burden when the coverage ends, especially in cases where fertility is preserved but the prognosis for future use is unclear.

*"I think that if I had children, I would try the natural method first, unless there's an issue on the other side... But there's also this thought that maybe the sperm I froze earlier isn't as healthy, given I was sick at the time... I imagine if it works,*

*it's viable and normal, but still..."* (Fabien, 29, Hodgkin's lymphoma at 28, single, no children, sperm cryopreserved pre-chemotherapy, no post-cancer spermogram)

This statement reflected the lack of knowledge among respondents who were not yet engaged in a parental project about the potential impact of cancer and treatment on the quality of their cryopreserved gametes. Some questioned whether their stored sperm would be viable for future conception, especially given the possibility of chemotherapy affecting sperm quality.

*"As much as you are told 'you absolutely must preserve,' there's still confusion. For instance, once you've banked your sperm, do you know how the process works? In the end, it was only by researching online that I understood the difference between an ART center and a sperm cryobank. It's still a bit of a grey area. It's not easy to understand that a sperm cryobank is just for storage, not for assisted reproductive treatment."* (Pierre, 28, Hodgkin's lymphoma at 27, sperm cryopreserved before chemotherapy, in a relationship, childless, azoospermia post-cancer, planning for parenthood)

This excerpt illustrates the confusion respondents faced due to the lack of information about their fertility status and the future use of cryopreserved gametes. In many cases, participants expressed uncertainty about their options for future parenthood, the impact of cancer on their fertility, and the complex process of using ART in the future.

### 3. Palliating infertility consequences in the context of a parental project: Beyond the sole issue of procreation after cancer

Among the interviewees who had undergone fertility preservation (FP), some were either planning to have children or had already become parents by the time of the interview. For these individuals, FP was a positive step in their post-cancer pathway, helping them move forward in various ways and at different stages of their recovery.

Among them, Robin's testimony illustrates a situation specific to men who, after cancer, experienced infertility but were able to preserve their fertility and later engaged in a parental project:

*"I feel like the medical profession saved me. To save me, they had to sacrifice certain functions... but I accepted that. Ultimately, I'm fortunate to have sperm cryopreserved, which allows me to have children without needing to rely on an anonymous donor."* (Pierre, 28, Hodgkin's lymphoma at 27, sperm cryopreserved before chemotherapy, in a relationship, childless, azoospermia post-cancer, planning for parenthood)

For men, known or presumed infertility cancer-induced can be effectively addressed through the use of cryopreserved sperm. This option provides reassurance, ensuring the possibility of future procreation and maintaining a genetic link to any future children.

The experience of infertility among the women interviewed, particularly those with a parental project, was more complex. The "reparative" aspect of using cryopreserved eggs or ovarian tissue added layers of difficulty. One such example is Mounia, 34, who was diagnosed with Hodgkin's lymphoma at 30. She underwent ovarian tissue cryopreservation prior to chemotherapy. At the time of the interview, Mounia had been in remission for three years and had recently undergone an ovarian tissue autograft that restored her endocrine function. She and her partner, who had been married for two and a half years, were actively trying to conceive with IVF:

*"What hurt me was that, during that whole period, I didn't receive hormone treatment. I don't know if it was a choice or if there were other options. I didn't have periods at all. It felt like I wasn't really a woman anymore; I had nothing. But when I came here [to the ART clinic], Dr. X gave me hope. She was the one who really cared for me. When I started my*

*first hormone treatment, and my periods came back, I was so emotional—I thought my ovaries had come back to life. I cried tears of joy."* (Mounia, 34, Hodgkin's lymphoma at 30, married with a child project, premature ovarian insufficiency (POI) after treatment, undergoing ovarian tissue transplantation)

In Mounia's case, as in several others, infertility due to POI (Premature Ovarian Insufficiency) is not resolved by the presence of cryopreserved gametes or tissue. The absence of menstruation, the climacteric syndrom such as hot flashes and vaginal dryness, sexual dysfunction, and other side effects induced by estrogen deficiency can complicate the path after cancer and cause additional challenges to conceive child:

*"I don't remember how long it lasted, but at some point, my periods stopped and then I started experiencing meno-pause symptoms—hot flashes, dry skin, dryness of the mucous membranes—classic signs of menopause. So, I saw Dr. X again, and she gave me everything I needed to counter those symptoms—hormones treatment, to restart my cycle, because I was still young, just 25. For her, it was out of the question for me to stay in this condition, especially since we wanted to have children. It was essential to find a solution."* (Laura, 31, Hodgkin's lymphoma at 24, in a relationship with two children aged 2 and 3, cryopreserved eggs prior to cancer treatment, conceived spontaneously despite POI diagnosis, undergoing Hormonal Replacement Therapy (HRT))

In Laura's case, as with Mounia's, the path to parenthood led them to consult an oncofertility specialist years after their cancer treatment had ended. It was within the framework of this parental project that solutions were offered to mitigate the effects of estrogen deficiency and infertility caused by POI since years.

## Discussion

Our research on fertility preservation in the context of cancer among young adults (AYA) revealed a variety of experiences, associated with different situations, including whether the respondents had a parental project or not, as well as their gender. This demonstrates that medical anticipation of risks through fertility preservation (FP) and access to assisted reproductive technologies (ART) does not eliminate all post-cancer difficulties and, in some cases, may even perpetuate them in ways that have not been considered.

The most positive experiences were linked to concrete medical responses to infertility within the framework of a parental project. In contrast, the most difficult experiences occurred in the absence of an actual parental project, for various reasons that we will elaborate upon.

On the one hand, many respondents, primarily women, who had not received counseling or FP before chemotherapy, reported a lack of information about the risk of infertility, which was often discovered too late, after cancer treatment had ended.

Even when FP was performed prior to treatment, the lack of information remained present in the study, but in a different way. Information regarding post-cancer infertility appeared in interviews only in the context of parental projects. In the absence of such a project, young single people were left with their questions, doubts, and sometimes even misconceptions about the status of their cryopreserved gametes, the conditions of their preservation, and how they could be used. Alongside this lack of information, testimonies showed what seemed to be a contradictory situation: the annual letter sent by the ART laboratory, asking what should be done with cryopreserved gametes, appeared as an excessive solicitation. It imposed a decision regarding a parental project at a time that did not match the life situation of many young singles who were not yet concerned with parenthood. The study questions this usual practice and its relevance in this particular context. The annual sending of this letter and the choices it presents were originally designed for supernumerary embryos, intended for people already parents who had to decide whether to pursue a second parental project or end it [41]. However, the specific situation of young single people in their twenties, who still consider themselves too young to think about having children, is not considered. Moreover, the study shows that this letter often arrives at a time when survivors wish

to distance themselves from their cancer experience, sometimes even contemplating the cessation of oncology follow-up. This desire is annually thwarted by the receipt of this letter, which directly reminds them of their cancer experience.

Additionally, the study reveals a lack of personalized counselling, particularly in addressing some respondents' misconceptions. Documentation via mail or email could be sent several months after the completion of treatment to explain how gamete cryopreservation works, the conditions for cryopreservation, and how these gametes may be used. It would also be beneficial to invite patients to contact reproductive specialists or the gamete cryopreservation center for further informations, if desired. Such a system could replace the annual reminder letters regarding the desired use of cryopreserved gametes, whose frequency does not align with the life situation of many young adults after cancer. These letters could be spaced out over time, for example, every 3–5 years, with a reminder about the possibility of making an appointment for more information.

On the other hand, our study showed that, beyond the issue of maintaining procreative capacity through FP, there are other concerns related to the post-cancer period, particularly among women, notably symptoms induced by premature ovarian insufficiency (POI), which are poorly managed and sometimes insufficiently compensated by hormone replacement therapy (HRT) (loss of libido, vaginal dryness, absence of menstruation, joint problems, hot flashes, etc.). A complementary letter could be sent to these women after their cancer treatment to offer them personalized gynecological follow-up, even if they are not involved in a parental project.

These results raise important questions about the adequacy of information, counseling, and care regarding endocrine function and fertility after cancer. While fertility preservation (FP) offers a "restorative" solution by anticipating the risk of post-cancer infertility, oncologists still do not sufficiently consider gender-specific issues, particularly the broader impact of premature ovarian insufficiency (POI) in women, which extends beyond the inability to conceive.

## Conclusion

Our research looked at the experiences of fertility preservation counselling in a cohort of AYA cancer survivors, taking into account a variety of situations: having been able to cryopreserve gametes or not, being still fertile after cancer or not, being involved in a child project or not. It thus makes it possible to take into account a variety of experiences of FP and to reveal the existence of relationships to this technical possibility that are more complex than the perspective of a simple "return to normality" through the maintenance of procreative capacities in the framework of a child project. FP, which is sometimes present outside of parental projects, gives rise to a set of questions that are sometimes disturbing, untimely and unwelcome. They are the result of a lack of informations and of taking into account the specific circumstances in which those involved in post-cancer FP sometimes find themselves: single and the lack of desire for a child, the desired distance from the cancer experience and the medical field, and the effects of the POI in women. We observed a need to develop information resources for AYA cancer survivors. Documentation by mail, sent several months after the end of the treatment, allowing to solicit oncofertility care staff if needed, would answer the numerous difficulties revealed, mainly due to a lack of information and consideration of the specificities of AYA after cancer.

## Acknowledgments

We thank all the women and men who agreed to participate in this study. We would like to thank all the staff who helped us in the recruitment of the participants, in particular the secretaries of the reproductive department and the AYA department of the oncofertility clinic where the study mainly took places.

## Author contributions

**Conceptualization:** Manon Vialle, Anne-Déborah Bouhnik, Julien Mancini, Blandine Courbiere.

**Data curation:** Manon Vialle.

**Formal analysis:** Manon Vialle.

**Funding acquisition:** Manon Vialle, Anne-Déborah Bouhnik, Julien Mancini, Blandine Courbiere.

**Investigation:** Manon Vialle.

**Methodology:** Manon Vialle.

**Project administration:** Manon Vialle.

**Resources:** Manon Vialle, Jacqueline Saias-Magnan, Anne Dezamis, Catherine Guillemain, Blandine Courbiere.

**Software:** Manon Vialle.

**Supervision:** Manon Vialle.

**Validation:** Manon Vialle.

**Visualization:** Manon Vialle.

**Writing – original draft:** Manon Vialle.

**Writing – review & editing:** Manon Vialle, Jacqueline Saias-Magnan, Anne Dezamis, Catherine Guillemain, Anne-Déborah Bouhnik, Julien Mancini, Blandine Courbiere.

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
