## [Decision Letter · Decision Letter 0]

2 Feb 2025

PONE-D-24-56910Exploring Fertility Preservation in AYA Cancer Survivors: Information Needs and Post-Cancer ChallengesPLOS ONE

Dear Dr. Vialle,

Thank you for submitting your manuscript to PLOS ONE. After careful consideration, we feel that it has merit but does not fully meet PLOS ONE’s publication criteria as it currently stands. Therefore, we invite you to submit a revised version of the manuscript that addresses the points raised during the review process.

We look forward to receiving your revised manuscript.

Kind regards,

Sayyed Mohammad Hadi Alavi

Academic Editor

PLOS ONE

Journal Requirements:

 “MV

INCA SHS-E-SP N°2017-137

The Institut National du Cancer (INCa)

URL: https://www.e-cancer.fr/Institut-national-du-cancer/Appels-a-projets/Appels-a-projets-resultats/SHS-RISP-2024

The funder didn't play any role”

Additional Editor Comments:

Thanks for your submission. Reviewers suggested your submission for publication, however there are some comments that needs consideration (very importantly study protocol from Reviewer #1). Reviewer #2 asked to remove a data and perform another stat analysis. Please consider this point in your revision. Very best regards, SMH Alavi, AE.

Reviewers' comments:

Reviewer's Responses to Questions

**Comments to the Author**

1. Is the manuscript technically sound, and do the data support the conclusions?

Reviewer #1: Yes

Reviewer #2: Yes

2. Has the statistical analysis been performed appropriately and rigorously? 

Reviewer #1: No

Reviewer #2: N/A

3. Have the authors made all data underlying the findings in their manuscript fully available?

Reviewer #1: Yes

Reviewer #2: Yes

4. Is the manuscript presented in an intelligible fashion and written in standard English?

Reviewer #1: Yes

Reviewer #2: Yes

5. Review Comments to the Author

Reviewer #1: I have thoroughly reviewed the manuscript titled “Exploring Fertility Preservation in AYA Cancer Survivors: Information Needs and Post-Cancer Challenges.” The authors employed sociological methodologies, including longitudinal interview analysis and cross-sectional analysis, with the aim of exploring the experiences of adolescent and young adult (AYA) cancer survivors regarding fertility preservation following cancer treatment. The study revealed significant disparities in participants' experiences related to fertility, such as a lack of attention to the risks of infertility, a disconnect between fertility preservation and pregnancy, and gaps in information and care. It underscores the insufficient focus on the specific issues faced by AYA cancer survivors, calling for the development of more targeted educational resources and support systems to address the unique needs of this population. This is a fascinating study; however, several aspects require refinement prior to publication:

1.The recruitment methods employed in the study provide assurance of sample diversity. Nevertheless, it is advisable to elaborate on the criteria and processes for sample selection in the methodology section, particularly regarding how the representativeness of the sample was ensured.

2.Regarding the interview process with participants, it would be beneficial to provide additional details about the design of interview questions and the methods of data analysis to enhance the transparency and reproducibility of the research. For instance, it would be pertinent to address the operational details of fertility preservation for both males and females, and whether participants had a certain level of understanding of these processes; it is known that the fertility preservation methods for females can be somewhat invasive compared to the sperm freezing techniques available for males.

3.In the results section, it is recommended to present the specific experiences and challenges faced by participants in fertility preservation more clearly, possibly utilizing quantitative data (such as participant proportions) and appropriate statistical methods to supplement the analysis and enhance credibility.

4.The discussion and conclusion sections should propose specific recommendations for clinical practice, particularly on how to improve support for fertility preservation among AYA cancer survivors, as well as directions for social or policy enhancements, while summarizing the primary findings of the study.

5.It is advisable to further emphasize in the ethical statement how participants' privacy and data security will be safeguarded.

Reviewer #2: The authors present a cross-sectional study including 31 adolescents and young adults cancer survivors, regarding their experiences of fertility preservation at the moment of cancer diagnosis.

The work is well presented and the conclusions are sound. However, a few comments should be addressed before considering this manuscript for publcation:

1. At the end of the Introduction section:

"This study, based on a qualitative sociological survey, aims to explore AYA survivors' post

cancer experiences in three key scenarios: 1) when the risk of infertility was not anticipated

and fertility preservation was not offered; 2) when fertility preservation was undertaken but

the AYA survivors do not yet have plans for parenthood; and 3) when fertility preservation

was implemented as part of an ongoing parental project."

It would have also been interesting to perform the same research in another scenario/group that we often find in the clinical practice 4) patients that when cancer is diagnosed are offered a fertility preservation treatment, but decide not to do it.

2. In the Material and Methods Section (page 5):

"Three cancer types were selected for inclusion: breast cancer, testicular cancer, and malignant hematological diseases".

Why other types of cancers with potentially gonadotoxic chemotherapy were not included (e.g. colon cancer)?

3. Results section (page 7):

"One participant, a woman diagnosed at 13 years old, was younger than our initial age

criterion"

This patient should be removed from the study, as she does not fulfill the previously cited inclusion criteria.

4. Results section (page 7):

"The participants' fertility statuses varied: some were still fertile (based on medical exams or

spontaneous pregnancies), others were infertile (confirmed by medical tests), and some had an

unknown fertility status (due to lack of medical examination or attempts to conceive)."

Fertility is a consequence of many factors. Thus, there are very few exams that can robustly confirm someone to be fertile or infertile. These exams should be stated.

5. Results: Patients names or personal data should not appear on the manuscript. Either the names are not real, and this deserves a comment in the Methods section, or they should be replaced by other data.

6. PLOS authors have the option to publish the peer review history of their article (what does this mean? ). If published, this will include your full peer review and any attached files.

**Do you want your identity to be public for this peer review?** For information about this choice, including consent withdrawal, please see our Privacy Policy .

Reviewer #1: No

Reviewer #2: No

---

## [Author Response · Author response to Decision Letter 1]

11 Apr 2025

Academic Editor:

-> Checked

“MV

INCA SHS-E-SP N°2017-137

The Institut National du Cancer (INCa)

URL: https://www.e-cancer.fr/Institut-national-du-cancer/Appels-a-projets/Appels-a-projets-resultats/SHS-RISP-2024

The funder didn't play any role”

-> The funders had no role in study design, data collection and analysis, decision to publish, or preparation of the manuscript.

-> We understand the importance of data sharing in ensuring transparency and replicability in research. However, in this case, we have opted for a restriction due to the sensitive nature of the data and our ethical commitments to the research participants. The raw data in our study consist of qualitative interviews that, even when pseudonymized, contain personal narratives that could compromise participant confidentiality. As part of our ethical protocol approved by the ethics committee, we guaranteed participants anonymity and the confidentiality of their data. Additionally, the protocol mandates the destruction of the interviews 5 years after analysis to further protect participant privacy.

Here are the contact details of the Aix-Marseille University ethics committee to which data requests can be sent. This is the committee that validated the ethical and methodological protocol for this research. Contact: audrey.janssens@univ-amu.fr

Although we are unable to share the raw interview data, we believe that the methodology and tools used ensure the reproducibility of our study. To this end, we have expanded on the presentation of the interview guide in the article (page 6) and the thematic analysis of the interviews (page 9). This will provide other researchers with the information they need to understand and, if necessary, reproduce our methodology and compare their results.

-> Following comments from reviewers, one reference has been added and another has been moved, as indicated below in the response to reviewers.

Reviewer #1:

1.The recruitment methods employed in the study provide assurance of sample diversity. Nevertheless, it is advisable to elaborate on the criteria and processes for sample selection in the methodology section, particularly regarding how the representativeness of the sample was ensured.

-> Added page 4: “In the medical reproduction department, an on-site presence, combined with the help of the department's secretaries and doctors, enabled this research to be systematically presented to eligible patients, who were systematically offered an interview, to be conducted at a later date if they so wished. In the oncology department, in partnership with the AJA department, an email presenting the research project and a call for participation were sent out to their email database. These two recruitment sources were the main ones, and in addition, word of mouth and the snowball effect helped expand recruitment beyond the locality of the two main departments, allowing for the consideration of any local specificities in the care provided. These recruitment methods and channels combined ensured a greater diversity of respondent profiles in terms of place of residence and care.”

2.Regarding the interview process with participants, it would be beneficial to provide additional details about the design of interview questions and the methods of data analysis to enhance the transparency and reproducibility of the research. For instance, it would be pertinent to address the operational details of fertility preservation for both males and females, and whether participants had a certain level of understanding of these processes; it is known that the fertility preservation methods for females can be somewhat invasive compared to the sperm freezing techniques available for males.

-> Regarding the design of interview questions, added page 6: “The first part of the interview guide focused on sociodemographic questions (age, gender, marital status, education, profession). The next sections were structured chronologically. A second part addressed the participants' personal, marital, and professional history up until the cancer diagnosis, as well as the possible emergence of a desire for a child during this period. The third part dealt with the cancer diagnosis, its treatment, experiences with medical care, including fertility-related issues, as well as the impacts and concerns in professional, social, family, and marital life. The fourth part of the interview focused on experiences after the completion of cancer treatment. Questions revisited the potential consequences of cancer and its treatments afterward, regarding fertility status (whether gamete preservation was performed or not, and whether fertility was preserved after treatment, if that information was known), and the experiences related to these aspects. Other questions then addressed the marital history, professional situation, desire for children after the end of treatment, and any parental project that was being considered, underway, or completed.”

-> Regarding the methods of data analysis to enhance the transparency and reproducibility of the research, added page 9: “Among the themes emerging from the interviews conducted, we find discourse on the proposal for fertility preservation (whether it was offered or not, when, by whom and how); the experience of fertility preservation when it was performed; the aftermath of treatment following the fight against cancer; the short-, medium-, and long-term consequences of treatments; the ignorance and late discovery of the effects on fertility; the misunderstanding of post-cancer difficulties on the part of relatives; the discovery of the premature ovarian insufficiency for women; experiences of infertility, whether real or assumed; the lack of information on the use of cryopreserved gametes and post-cancer infertility; the desire for children or not; the parental project; and the difficulties faced in assisted reproduction treatments (ART). Within each of these themes, several other categories emerged. In this paper, we will focus on certain data and concentrate our analysis on the following points.”

-> Regarding the operational details of fertility preservation for both males and females, added page 10: “. In addition, the greater number of men than women using FP in our sample [Table 1] can be explained by a point that is well documented in the literature, namely that gamete retrieval is easier for men than for women in terms of time, technique and invasiveness for the body [40]. As regards the use of FP by the women in our sample, the situations are indeed more varied and complex.” Here we also referred to the following added paper:

[40] Perachino M, Massarotti C, Razeti MG, Parisi F, Arecco L, Damassi A, et al. Gender-specific aspects related to type of fertility preservation strategies and access to fertility care. ESMO Open, Volume 5, Supplement 4, 2020, e000771, ISSN 2059-7029, https://doi.org/10.1136/esmoopen-2020-000771.

3.In the results section, it is recommended to present the specific experiences and challenges faced by participants in fertility preservation more clearly, possibly utilizing quantitative data (such as participant proportions) and appropriate statistical methods to supplement the analysis and enhance credibility.

We have made every effort to indicate, as precisely as possible, the number of interviews corresponding to each situation within our sample in the results section. While we are able to provide exact counts, we cannot present proportions or statistical analyses due to the qualitative methodology used. Indeed, the aim of a qualitative study is not to be quantitatively representative, but rather to provide in-depth insight into lived experiences, striving for as much diversity as possible. And while these experiences may be representative of real and commonly shared situations, they are not quantitatively representative. Therefore, calculating proportions or statistics based on this sample would not be scientifically valid. The primary value of the qualitative scientific approach used here lies in its ability to deeply explore and better understand the lived experiences related to the care—or lack thereof—of fertility issues after cancer.

4.The discussion and conclusion sections should propose specific recommendations for clinical practice, particularly on how to improve support for fertility preservation among AYA cancer survivors, as well as directions for social or policy enhancements, while summarizing the primary findings of the study.

It seems to us that this requirement is met, in particular through the inclusion of the following recommendations after the summary of our results:

-> Page 21: “Documentation via mail or email could be sent several months after the completion of treatment to explain how gamete cryopreservation works, the conditions for cryopreservation, and how these gametes may be used. It would also be beneficial to invite patients to contact reproductive specialists or the gamete cryopreservation center for further informations, if desired. Such a system could replace the annual reminder letters regarding the desired use of cryopreserved gametes, whose frequency does not align with the life situation of many young adults after cancer. These letters could be spaced out over time, for example, every 3 to 5 years, with a reminder about the possibility of making an appointment for more information.

-> Page 22: “A complementary letter could be sent to these women after their cancer treatment to offer them personalized gynecological follow-up, even if they are not involved in a parental project.”

-> Page 22-23: “We observed a need to develop information resources for AYA cancer survivors. Documentation by mail, sent several months after the end of the treatment, allowing to solicit oncofertility care staff if needed, would answer the numerous difficulties revealed, mainly due to a lack of information and consideration of the specificities of AYA after cancer.

5.It is advisable to further emphasize in the ethical statement how participants' privacy and data security will be safeguarded.

-> Added, page 7: “All the data collected was kept solely by the research investigator on a secure password-protected area. The data stored was all anonymized, and the ethical protocol approved by the committee stipulates that it must be destroyed 5 years after the end of the study.”

Reviewer #2:

1. At the end of the Introduction section:

"This study, based on a qualitative sociological survey, aims to explore AYA survivors' post

cancer experiences in three key scenarios: 1) when the risk of infertility was not anticipated

and fertility preservation was not offered; 2) when fertility preservation was undertaken but

the AYA survivors do not yet have plans for parenthood; and 3) when fertility preservation

was implemented as part of an ongoing parental project."

It would have also been interesting to perform the same research in another scenario/group that we often find in the clinical practice 4) patients that when cancer is diagnosed are offered a fertility preservation treatment, but decide not to do it.

-> I agree with the relevance of studying this fourth category. However, the three proposed categories stem from themes that emerged during the data analysis (see the added section on analysis themes, p.9. Since this particular situation did not arise within our sample, we were not able to study it.

2. In the Material and Methods Section (page 5):

"Three cancer types were selected for inclusion: breast cancer, testicular cancer, and malignant hematological diseases".

Why other types of cancers with potentially gonadotoxic chemotherapy were not included (e.g. colon cancer)?

-> Added page 5: “We focused on three types of cancer in order to maintain a certain level of homogeneity in both the types of cancers and the oncology and oncofertility care pathways, allowing us to better identify the specificities arising within these different journeys. We chose to concentrate on hematologic cancers, which particularly affect young people, as well as two more gender-specific cancers—testicular cancer and breast cancer—one of which (breast cancer) can be hormone-dependent in women, in order to take into account how gender and hormonal factors may influence the experience of illness, fertility preservation, and reproductive decisions.”

3. Results section (page 7):

"One participant, a woman diagnosed at 13 years old, was younger than our initial age

criterion"

This patient should be removed from the study, as she does not fulfill the previously cited inclusion criteria.

We understand that it may have been unclear to state that our inclusion criterion was being between 15 and 35 years old at the time of cancer diagnosis, while including a participant who was 13 at the time. However, we believe the rationale for her inclusion remains justified: she falls within certain definitions of the AYA group, and the issues she faced as an adult are central to our study and shared by the other participants. In this context, we believe it is more accurate to clarify that our sampling criterion ranged from ages 13 to 35. To reflect this, we have moved one reference and modified the following section:

-> added pages 4-5: “The AYA category was defined broadly, ranging from 13 to 35 years old, in line with re

---

## [Editor Report · Decision Letter 1]

16 Apr 2025

Exploring Fertility Preservation in AYA Cancer Survivors: Information Needs and Post-Cancer Challenges

PONE-D-24-56910R1

Dear Dr. Vialle,

We’re pleased to inform you that your manuscript has been judged scientifically suitable for publication and will be formally accepted for publication once it meets all outstanding technical requirements.

Kind regards,

Sayyed Mohammad Hadi Alavi

Academic Editor

PLOS ONE

Additional Editor Comments (optional):

Thanks for your revision that includes the reviewers' comments.

Your MS may be reviewer the Expert Staff for Ethics and for MS formatting (Abstract and References).
---

## [Editor Report · Acceptance letter]

PONE-D-24-56910R1

PLOS ONE

Dear Dr. Vialle,

I'm pleased to inform you that your manuscript has been deemed suitable for publication in PLOS ONE. Congratulations! Your manuscript is now being handed over to our production team.

Kind regards,

on behalf of

Dr. Sayyed Mohammad Hadi Alavi

Academic Editor

PLOS ONE